# Catalytic Conversion of Xylose to Furfural by *p*-Toluenesulfonic Acid (*p*TSA) and Chlorides: Process Optimization and Kinetic Modeling

**DOI:** 10.3390/molecules26082208

**Published:** 2021-04-12

**Authors:** Muhammad Sajid, Muhammad Rizwan Dilshad, Muhammad Saif Ur Rehman, Dehua Liu, Xuebing Zhao

**Affiliations:** 1Faculty of Materials and Chemical Engineering, Yibin University, Yibin 644000, China; 2Institute of Applied Chemistry, Department of Chemical Engineering, Tsinghua University, Beijing 100084, China; dhliu@tsinghua.edu.cn; 3Department of Chemical Engineering, University of Gujrat, Gujrat 50700, Pakistan; 4Institute of Chemical Engineering and Technology, University of the Punjab, Lahore 54590, Pakistan; rizwandilshad@gmail.com; 5Department of Chemical Engineering, Khwaja Fareed University of Engineering and Information Technology, Abu Dhabi Road, Rahim Yar Khan 54000, Pakistan; saifrehman1224@yahoo.com

**Keywords:** catalysis, chlorides, furfural, kinetics, lewis acid, *p*-toluenesulfonic acid, xylose

## Abstract

Furfural is one of the most promising precursor chemicals with an extended range of downstream derivatives. In this work, conversion of xylose to produce furfural was performed by employing *p*-toluenesulfonic acid (*p*TSA) as a catalyst in DMSO medium at moderate temperature and atmospheric pressure. The production process was optimized based on kinetic modeling of xylose conversion to furfural alongwith simultaneous formation of humin from xylose and furfural. The synergetic effects of organic acids and Lewis acids were investigated. Results showed that the catalyst *p*TSA-CrCl_3_·6H_2_O was a promising combined catalyst due to the high furfural yield (53.10%) at a moderate temperature of 120 °C. Observed kinetic modeling illustrated that the condensation of furfural in the DMSO solvent medium actually could be neglected. The established model was found to be satisfactory and could be well applied for process simulation and optimization with adequate accuracy. The estimated values of activation energies for xylose dehydration, condensation of xylose, and furfural to humin were 81.80, 66.50, and 93.02 kJ/mol, respectively.

## 1. Introduction

Furfural is a high-value furan aldehyde, with a wide range of industrial uses, especially in the production of pharmaceuticals, polymers, resins, solvents, fine chemicals, fuel additives, and biofuels [1,2,3,4]. Xylose is an encouraging feedstock for furfural synthesis because of the process simplicity and product selectivity [5,6]. 

Numerous reports have described the different aspects of xylose dehydration to furfural [1,3,5]. Ample interest in the development of techno-economical processes has steered the evaluation of comprehensive reaction kinetics in order to illuminate the detailed reaction mechanism [7]. It has been observed that catalytic conversion of xylose resulted in the simultaneous production of furfural and some miscellaneous products (Scheme 1). 

Mineral acids [8,9], modified solid acids [10], metal oxides [11], and some organic acids [12], have been used for the production of furfural from xylose by employing water, organics, ionic liquids, and their combinations as solvents. Generally, mineral acids are applied in organic synthesis because of the higher process yields, and heterogeneous solid acids are used because of their high recycling efficiencies [9,13]. Besides these advantages, mineral acids are less selective owing to associated environmental concerns [14]. On the other hand, solid acid catalysts typically lack recyclability after the first cycle due to fouling and preferred adsorption of stable process intermediates; hence these are less efficient for mass production, especially for biomass-derived components [15,16]. Ionic liquids were found to be relatively efficient reaction media having integrated functions of catalysts as well as solvents [17]. Nevertheless, high cost, low recovery, and poor recycling are substantial hindrances for their industrial applications [18]. Interest in the replacement of mineral acids by organic acids as the catalyst is increasing to overcome the stubborn problems of environmental pollution [19]. Hence, organic acids have been acknowledged as prospective catalysts for the dehydration of sugars. Additionally, only a 50% furfural yield has been achieved in the industrial mineral acid-catalyzed furfural production process [20]. Hazardous waste problems due to HCl and/or H_2_SO_4_, and high-energy requirements (the reaction temperature is typically 200 °C) are some observed drawbacks. Hence, an alternative method with mild reaction conditions, improved energy efficiency, and decreased environmental problems are needed. Biocatalysis is also an alternative non-toxic route to make the desired product under milder reaction conditions; however, slow reaction rates and delicate process control are vital for an excellent yield [21,22,23]. Meanwhile, organic acids have low corrosion impact and are reasonably selective considering the process economics and rate of reaction [24]. Moreover, thermal decomposition of organic acids yield relatively less toxic and recyclable components like HCOOH, CO_2_, and fumaric acid avoiding waste stream contamination [24]. Thus, organic acids can be considered as potential chemical catalysts having the benefits of homogenous catalysis as well as relatively low environmental toxicity.

Recently, the kinetics of xylose dehydration have been discussed applying heterogeneous catalysts [25,26] and homogenous catalysts, including mineral acids [27] and organic acids [12]. Although biphasic solvent systems are now widely used, kinetic investigations of xylose dehydration were mostly studied in the aqueous phase at high temperature and high-pressure [28].

Voluminous attention have been paid to mineral acids to be used in homogenous catalysis for xylose conversion to furfural. HCl was used for the biphasic kinetic analysis of xylose dehydration, applying microwave heating [29]. Results indicated a faster rate of xylose decomposition to humin than xylose dehydration to furfural. Shifting the reaction medium from monophasic to biphasic by the addition of an organic solvent improved the furfural yield from 30% to 85%. However, it was noted that the reaction kinetics remained unchanged in both reaction media either microwave irradiation or conventional heating were applied [29]. H_3_PO_4_ and H_2_SO_4_ were also used as homogeneous mineral acid catalysts [30,31]. Similar benefits are also available with organic acids, but these have not been thoroughly investigated. The use of organic acids may be the most attractive reaction system considering the furfural production along with integrated organosolv fractionation [12]. Kinetic dehydration of xylose employing formic acid was performed at high temperatures (130–200 °C). Similar to formic acid, acetic acid [32] and *p*-toluenesulfonic acid [25] are also equally efficient for dehydration reactions. However, a comprehensive systematic kinetic investigation of the process and process mechanism has not been performed.

It has been witnessed that the addition of chlorides can improve the process efficiency of sugars dehydration to furan aldehydes [27,33], and organic acids have the benefits of homogenous catalysis similar to mineral acids with low environmental toxicity [2,34]. *p*TSA, being solid at room temperature, has the advantage of easy recovery using established precipitation techniques [25]. Therefore, in this work, a combination of organic acids and chlorides was investigated using organic solvents for the eco-friendly production of furfural. The dependency of the product yield of different process parameters was optimized using kinetic modeling. The role of Lewis acid addition was calculated considering the maximum furfural yield. Additionally, the reaction system was optimized considering the kinetic parameters and maximum process yield. A comparative evaluation of the experimental results and corresponding kinetic data elucidated that CrCl_3_·6H_2_O is the best promoter for the *p*TSA-catalyzed conversion of xylose to produce furfural. The *p*TSA-CrCl_3_·6H_2_O catalyst is cheap and ecologically friendly as compared to mineral and solid acid catalysts. The proposed process thus has the potential to be operated as a tool for advanced optimization to achieve the maximize furfural yield from biomass-derived sugars.

## 2. Results and Discussion

### 2.1. Aqueous Phase Dehydration of Xylose into Furfural

Synthesis of furfural from xylose was investigated using optimized reaction conditions from our previous experiments (conversion of C_6_ sugars to HMF) and then again optimized [33,34]. Water is a preferred solvent because of process economics, ecological concerns, high sugar dissolving capacity, and versatile physio-chemical properties. Applying organic acids in pure form, a meager furfural yield was observed, as the applied temperature was 100 °C due to pressure constraints (Appendix A). Köchermann et al. studied the aqueous phase xylose conversion using sulfuric acid as a homogeneous catalyst [35]. Applying an aqueous phase high-pressure reactor, the obtained furfural yield was 36.3% in 15 min with 91.8% xylose conversion at 200 °C. A 52.0% of the furfural yield was achieved by increasing the temperature to 220 °C, which showed the positive impact of higher temperatures [35]. High temperatures can be secured either by using a high-pressure reaction system or by using high boiling point solvents. Therefore, a higher pressure (6.0 MPa) was maintained to achieve a high temperature in the water medium [35]. Continuous provision of such a high pressure has economic and safety concerns, so we have fixed the pressure as atmospheric pressure considering the industrial safety and energy expenditure, so an increase in pressure is not a sustainable preference and hence was not elaborated.

### 2.2. Effect of Lewis Acid Addition

It has been observed that the addition of Lewis acid improved the isomerization and dehydration efficiency of the organic acid-catalyzed process when used in glucose dehydration reactions [33]. Similar results were also reported in the mineral acid-catalyzed xylose dehydration process as the aqueous phase HCl catalyzed furfural yield was increased to 38.7% from 28.8% with the addition of 13.5 mM CrCl_3_·6H_2_O at 145 °C [36]. Thus, different Lewis acids were added to investigate the synergy of organic acid and Lewis acid combinations. CrCl_3_·6H_2_O gives the highest furfural yield in an aqueous medium (Figure 1A), similar to published literature [14,36]. Concentration effects were investigated by applying 0.1 M to 0.6 M of CrCl_3_·6H_2_O and keeping all the other parameters constant. An increase in catalyst concentration was unswervingly reciprocated in the xylose conversion and furfural yield.

The maximum furfural yield of 30.05% was recorded with 82.10% xylose conversion at 0.5 M concentration of CrCl_3_·6H_2_O after this decreasing trend was observed (Figure 1B). These results directed the positive impact of Lewis acid addition on furfural yield. It has been proved that metal halides strongly affect the dehydration rate in an aqueous medium [37]. Xylose dehydration increased with a decrease in the water activity and water activity is decreased by the addition of halides in the following order Cl^−^ >Br^−^ > I^−^ [37]. When the effect of trivalent chlorides was considered, Cr gives the maximum efficiency following the order of CrCl_3_ > AlCl_3_ > InCl_3_ > FeCl_3_ [38]. Molecular dynamics simulations elucidated that CrCl_3_·6H_2_O produces Cr(H_2_O)_5_OH^2+^ when combined with water at pH ≤ 4 [39]. Cr(H_2_O)_5_OH^2+^ is the more reactive species and affects the dehydration of sugars positively. Therefore the synergetic effects of HCl-CrCl_3_·6H_2_O and H_3_PO_4_-CrCl_3_·6H_2_O gave the best results for the conversion of glucose to HMF [39,40]. Similar results were also obtained in our earlier findings employing a combined *p*TSA-CrCl_3_·6H_2_O catalyst [33]. In the same way, these results also illuminated that CrCl_3_·6H_2_O is a more active supporting catalyst than AlCl_3_ and NH_4_Cl. It is further speculated that side reactions of xylose, reaction intermediates, and produced furfural form unwanted side products that were responsible for the lower furfural yield in water medium [37,38]. To further check the effects of temperature and solvent medium on the product selectivity, water was replaced with organic solvents.

Kinetic analysis, as shown in Appendix A, also confirms the catalytic superiority of the *p*TSA-CrCl_3_·6H_2_O co-catalyzed process. The principal rate of reaction (*k*_1obs_) increased with the addition of Lewis acids in the order of CrCl_3_·6H_2_O > AlCl_3_ > NH_4_Cl > no Lewis acid. An increase in *k*_2obs_ and *k*_3obs_ was also perceived along with *k*_1obs_ with the addition of CrCl_3_·6H_2_O, which is also a confirmation of the active catalytic performance of the selected reaction system. It has been observed that trivalent chromium salts are highly efficient in aqueous as well as in aprotic solvent-based dehydration reactions [41]. Therefore, a gradual increase in *k*_1obs,_ with an increase in CrCl_3_·6H_2_O concentration was experienced (Appendix A).

The increase in the prime rate of reaction is similarly reciprocated by an increase in product yield. The change in *k*_2obs_ and *k*_3obs_ was not uniform. It can be witnessed from Appendix A, that *k*_1obs_ increased from 1.16 × 10^−3^ min^−1^ to 1.70 × 10^−3^ min^−1^ by changing the CrCl_3_·6H_2_O concentration from 0.4 M to 0.5 M and a similar increase was also noted in *k*_2obs_, whereas, contrarily *k*_3obs_ decreased from 1.14 × 10^−4^ min^−1^ to 5.21 × 10^−11^ min^−1^. Kinetic results showed the narrow variation of *k*_1obs_ and *k*_2obs,_ while a relatively high variation in *k*_3obs_ (10^−12^–10^−4^) was recorded. At low concentration, *k*_2obs_ were higher than *k*_1obs_; whereas, at high concentration, the observed rate constants were nearly equal (*k*_1obs ≈_
*k*_2obs_). This observation indicated the higher tendency of xylose decomposition to side products (denoted as humin) than furfural formation in water. Weingarten et al., also reported that the rate of aqueous phase xylose decomposition to side products (*k*_2obs_) was higher in the HCl-catalyzed process [29]. Results reported by Lamminpä ä et al., employing HCOOH as a catalyst also suggested a similar phenomenon [12]. The observed rates were nearly equal and calculated *k*_1obs_ and *k*_2obs_ were 6.85 × 10^−4^ min^−1^ and 2.28 × 10^−4^ min^−1^, respectively [12]. The calculated kinetic parameter in this study (Appendix A) are in good consistency with the reported literature values. The kinetic plots (Figure 1 and Appendix A) show a good agreement between the model’s predicted values and experimentally determined values. All the goodness of fit (*R*^2^) values is >0.95, which proves the accuracy of the developed model.

### 2.3. Conversion of Xylose to Furfural in Organic Medium

Optimized parameters were applied to check the quantitative effects of polar aprotic organic solvents (DMSO and DMF) on furfural yield. As indicated in Figure 2A, high product yield was observed in the DMSO solvent medium as compared to DMF and water solvent media. Quantitatively, the observed experimental yield order was DMSO > H_2_O > DMF. Surprisingly, the furfural yield in the aqueous phase (30.05%) was a little higher than in the DMF phase (29.74%). However, the reaction time needed in the aqueous phase (6 h) was longer than in the DMF phase (1 h); whereas, the maximum furfural yield was obtained in only 0.5 h when DMSO was used as a solvent medium. It has been noted that the use of water as a monophasic solvent medium for the conversion of xylose to furfural accelerates the side reactions which adversely affects the product selectivity [36,42]. Similar effects were also observed in our previous experiment of the conversion of hexoses to HMF [33,34]; this phenomenon further proved the preferred solvation property of DMSO toward furan aldehydes. Therefore, the furfural yield in theaqueous phase was lower as compared to DMSO solvent medium. It was also observed that a higher substrate concentration exceeded the product yield in homogeneous catalysis [43]. In this study, high xylose concentration may be a reason for this increase in yield. DMSO, an organic solvent, could solve this issue. DMSO was also the best organic solvent for the HMF production process from C_6_ sugars because of its steadiness even at high-temperature and superior solvation toward furan aldehydes [33,34]. It may be partly because DMSO can effectively introverted the furfural decomposition into unwanted side products [43].

Kinetic analysis of xylose conversion in organic medium has been performed similarly to the aqueous phase process. The kinetic curves of xylose, furfural, and humin concentrations are provided in Figure 3, and observed rate constants are listed in Table 1. The kinetic comparison proved the solvation superiority of DMSO. The highest *k*_1obs_ was observed when DMSO was used as a solvent medium and *k*_1obs_ increases in the order of DMSO > DMF > H_2_O. However, when the rate constants of organic solvents were compared, high *k*_1obs_, lower *k*_2obs,_ and *k*_3obs_ were found with DMSO than DMF (Table 1). The magnitude of *k*_1obs_ and *k*_2obs_ is larger than *k*_3obs_; therefore, furfural decomposition could be neglected under these process conditions. These results indicated that the rate constants in water media are far less than in the DMF medium, but the recorded product yield was a little higher in the water medium. This is because of the high proportional effect of the prime rate of reaction to undesired rate of reaction in the water medium than in the DMF medium. The data, as shown in Table 1 shiwed that a higher reaction selectivity (*S*_F_ = *k*_1obs_/*k*_2obs_) could be obtained in the DMSO medium than in DMF and water media. Therefore, DMSO was found to be the most promising solvent for this organic acid-catalyzed dehydration of xylose to furfural. Dehydration reactions are ominously influenced by the reaction temperature, and it is generally accepted that furfural moieties cannot undergo degradation reactions in the vapor phase [42].

The established reaction parameters (0.50 M xylose, 1.0 M *p*TSA, and 0.50 M CrCl_3_·6H_2_O) were then used to appraise the dependency of furfural yield and selectivity on temperature employing DMSO as a reaction medium. The process temperature was synchronized from 110 °C to 160 °C, and the results are presented in Figure 2B. Kinetic data are presented in Table 2 and the corresponding kinetic plots in Appendix A. The results proved that the established kinetic model can successfully explain the process kinetics of this reaction system. The developed kinetic plots revealed the significant impact of temperature on furfural selectivity. It can be observed from the results that furfural yield increased with an increase in temperature from 110 °C to 120 °C. However, a further increase in temperature from 120 °C to 160 °C affects inversely, reducing the furfural yield. This shoddier furfural yield may be attributed to the higher rate of xylose decomposition to humin than xylose dehydration to furfural [42]. The rate of furfural decomposition (*k*_3obs_) remains substantially low throughout the experiment. This can also be wittnessed by the concentration profiles showing a continuous increase in the furfural concentration followed by a minor decline next to a peak value. However, a high rate of furfural decomposition (*k*_3obs_) is observed at high reaction temperatures, especially beyound 140 °C. All the rate constants improved with an increase in temperature, and *k*_2obs_ were higher than *k*_1obs_ at all temperatures except 130 °C. The objective reaction (conversion of xylose to furfural) had a smaller rate constant (*k*_1obs_) than humin formation (*k*_2obs_) but the difference was only 5%, so it can be assumed that xylose has an equal rate of dehydration and decomposition at all temperatures in the DMSO medium. The humin formation was lowest at 120 °C, which was increased with the temperature rise. Therefore, the magnitude of the rate of xylose decomposition to humin (*k*_2obs_) was 0.022 min^−1^ at 120 °C, and it increased gradually and the highest value of 0.2948 min^−1^ was observed at 160 °C. Correspondingly, a change in color from light yellow to dark brown is observed which proved the increased byproduct formation in the reaction mixture. A kinetic study of xylose dehydration in high-pressure liquid water also reported a rise in *k*_2obs_ with increased temperatures [44]. A similar increase in rate constants with temperature escalation has also been reported in the literature [12,32,41]. The magnitude of *k*_1obs_ and *k*_2obs_ changed from 0.00322 min^−1^ and 0.00128 min^−1^ to 0.0356 min^−1^ and 0.0297 min^−1^, respectively, with a temperature rise from 180 °C to 220 °C [44]. Hence, it can be validated that temperature strongly stimulates the product selectivity in the *p*TSA-CrCl_3_·6H_2_O catalyzed process of xylose dehydration to furfural in the DMSO medium, and a careful balance is required between the objective reaction and side reactions to achieve a high product selectivity.

Metal chlorides were found equally efficient in water as well as in organic solvent media. However, a high concentration of Lewis acids may result in an additional financial burden. Therefore, the impact of Lewis acid contents on furfural selectivity was further optimized in the DMSO medium, as the purpose was to check the process efficiency under reduced concentration conditions. Hence, the CrCl_3_·6H_2_O level was varied between 0.1 M to 0.5 M, and obtained results are presented in Figure 2C, and kinetic parameters are shown in Table 3 and Appendix A. Surprisingly, the results suggested an inverse relation between furfural yield and Lewis acid concentration, and the highest furfural yield of 52.60% was attained with 0.1 M CrCl_3_·6H_2_O concentration. Similar phenomena could be observed in the kinetic parameters (Table 3). The principal rate constant (*k*_1obs_) increases inversely with CrCl_3_·6H_2_O concentration. The magnitude of *k*_1obs_ and *k*_2obs_ increased to 0.0508 min^−1^ and 0.0542 min^−1^ from 0.033 min^−1^ and 0.047 min^−1^, respectively, by decreasing the CrCl_3_·6H_2_O concentration from 0.5 M to 0.1 M. This increase in product selectivity with a decrease in system acidity proves the negative impact of high acidity. By decreasing CrCl_3_·6H_2_O from 0.5 M to 0.1 M, a proportional growth of 54.0% and 15.1% in *k*_1obs_ and *k*_2obs_ was noted, respectively. This significant difference in a proportional increase in rate constants (*k*_1obs_ than *k*_2obs_) was straightly reciprocated in furfural yield.

The reaction selectivity (*S*_F_ = *k*_1obs_/*k*_2obs_) also increased with a decrease in CrCl_3_·6H_2_O concentration and the highest value of 0.94 is observed at the lowest concentration (Table 3). The magnitude of *k*_3obs_ remains in the range of 10^−4^, so it can be concluded that a change in Lewis acid concentration does not affect the furfural decomposition reaction appreciably and furfural decomposition remains very low at all proportion. Kinetic plots, as shown in Appendix A, verified this hypothesis, which is shown by constant furfural concentration after reaching a peak value. Thus, it can be established that the principal reaction of xylose dehydration into furfural in the DMSO solvent medium increased with a decrease in CrCl_3_·6H_2_O concentration in contrast to the aqueous phase reaction.

### 2.4. Comparison of Different Organic Acids

The impact of different organic-Lewis acid catalytic combinations in the DMSO medium was further explored. The kinetic profiles of xylose, furfural, and byproducts (humin) are presented in Figure 4, and the resultant parameters are enumerated in Table 4. The high *R*^2^ (≥0.95) again proved the accuracy of the developed kinetic model. The observed rate constants elucidated that the xylose conversion is in the sequence of *p*TSA > oxalic acid > maleic acid > succinic acid > malonic acid (Table 4). However, the maximum product obtained shows little deviation from this trend. The values of *k*_1obs_ were in the range of 10^−3^–10^−2^ min^−1^. Corresponding reaction selectivity for xylose conversion (*k*_1obs_/*k*_2obs_) was determined to be 0.94, 0.71, 0.52, 0.0.66 and 0.51, respectively. It has been observed the *p*TSA-CrCl_3_·6H_2_O catalyst obtained the maximum furfural yield in DMSO medium, similar to the aqueous phase process. Furfural yields obtained in the order of *p*TSA > oxalic acid> succinic acid > maleic acid > malonic acid.

The reaction mechanism can be explained assuming a 1st order conversion of reactant X to product F, as X→F for xylose conversion having an apparent differential rate equation given by the following expression:(1)$$\frac{d\left[F\right]}{dt}=k_{R}\left[X\right]$$

Here *k*_R_ is the observed rate constant. As the reaction is catalytic and proceeded with the addition of homogenous catalysts; therefore, incorporating catalytic effect [H^+^] with quasi-stationary distribution, the kinetic model could be expressed as [34]:(2)$$\frac{d\left[F\right]}{dt}=k_{R2}^{}\frac{k_{R1}^{}}{\left(k_{R2}^{}+k_{R-1}^{}\right)}\left[X\right]\left[H^{+}\right]$$

From Equations (1) and (2), it is evident that *k*_R_ depends upon [H^+^] linearly with the following relationship:(3)$$k_{R}=\frac{k_{R1}^{}k_{R2}^{}}{k_{R2}^{}+k_{R-1}^{}}\left[H^{+}\right]$$

*p*TSA is a relatively strong organic acid with a *p*K_a1_ value of −2.8; hence complete dissociation can be assumed, whereas, for other organic acids, the aqueous phase [H^+^] can be estimated by the following equation [45]:(4)$$\left[H^{+}\right]\approx \sqrt{C_{Acid}K_{a1}}$$
here *C_Acid_* represents the acid catalyst concentration. Considering this relationship (Equation (4)), the [H^+^] of the used organic acid catalysts was calculated and plotted in Figure 5. The results demonstrated a good linear relationship (*R*^2^ = 0.9786) and prove the accuracy of the developed model.

The comparison of *p*K_a1_ indicated *p*TSA as the strongest acid (*p*K_a1_ = −2.8) among all the organic acids applied which results in its almost complete dissociation providing the highest [H^+^] in the reaction system. The rate of reaction is roughly directly proportional to proton concentration {[H^+^]} in acid-catalyzed reactions, as proved in Equation (3) (Figure 5). Furthermore, the synergetic effects of CrCl_3_·6H_2_O addition contribute positively to the sugars dehydration reactions. Hence, the high xylose conversion was achieved in only one hour whereas other organic acid catalysts required up to five hours to achieve similar results. These variations indicate that every mixed acid system do not have similar synergistic effect [33,39,40]. Therefore, the high xylose conversion may be attributed to the combined effects of *p*TSA-CrCl_3_·6H_2_O. Oxalic acid (*p*K_a1_ = 1.27), likewise exhibited relatively better catalytic efficacy for xylose dehydration and gave a furfural yield of 42.64%. Interestingly, succinic acid having the lowest acidity (*p*K_a1_ = 4.2) gives a higher product yield than maleic acid (*p*K_a1_ = 1.9) and malonic acid (*p*K_a1_ = 2.83). It has been proved that the correlation of [H^+^] and *k* is linear (Equation (10)); this anomaly can be explained by several reasons. Firstly, the synergy of succinic-CrCl_3_·6H_2_O positively affects the principal reaction (xylose dehydration to furfural), which is also confirmed by the high reaction selectivity (S_F_ = 0.66). Secondly, the synergetic effect may have changed the combined acidity at the reaction temperature. It has also been reported that numerous other factors were also contributing and partially influenced the dehydration mechanism [42,46]. Furthermore, the variation in *p*K_a1_ of the organic acids with reaction temperature and solvent medium should also be considered and a detailed investigation is further required.

The literature indicates that metal chlorides (FeCl_3_, AlCl_3_, and CrCl_3_) mostly give 30–45% furfural yield depending upon the system concentrations, in the absence of Brønsted acids [38]. Similarly, only a 33.8% furfural yield was achieved by employing CrCl_3_·6H_2_O as a sole catalyst. In this study, the high furfural yield with high sugar contents is credited to the synergetic effects of *p*TSA-CrCl_3_·6H_2_O catalyst in the DMSO solvent medium.

The substrate concentration effects were further examined by varying the initial xylose concentrations between 0.25–1.0 M while keeping all the other parameters constant (1.0 M *p*TSA, 0.10 M CrCl_3_·6H_2_O, 120 °C, and atmospheric pressure). The obtained results and fitted kinetic parameters are provided in the Appendix A. The time course concentration profiles indicate that there was no significant influence of initial xylose concentration on the conversion rates, while a minimal impact can be noted on the product yield. When the xylose concentration was changed from 0.25 M to 0.5 M, the furfural yield decreased from 53.1% to 52.6%. However, with increasing the initial concentration to 1.0 M, the furfural yield decreased to 42.4%. At a high concentration of xylose, the negative effect on furfural yield is due to multiple reasons [8,17]. Simultaneous decomposition of xylose and furfural to humin along with reactive intermediates are probable reasons. It can be observed from the concentration profiles (Appendix A) that xylose conversion is a relatively fast reaction, and more than 80% conversion was achieved in the first 15 min. However, the complete transformation was delivered after 2 h, whereas, the maximum furfural yield was attained in between 1–1.5 h. After achieving the maximum value, a slight decrease in furfural concentration can be observed. Furthermore, at high concentrations, some process intermediates remained unconverted, and a longer reaction time was required for their complete conversion. Probably, a non-uniform mass distribution at high concentration results in a decreased number of effective molecular collisions [7,47]. It has been observed that prolonged reaction time results in an increased rate of humin formation from furfural and other process intermediates [37]. Hence, the cumulative effect of high substrate concentration resulted in decreased product yield.

The relationship between temperature, the rate constant, and catalyst concentration could be elaborated using the modified form of the Arrhenius equation:(5)$$k=Ae^{\left(\frac{-E_{a}}{RT}\right)}C_{\mathrm{acid}}^{\alpha }$$
(6)$$\ln k=\ln A-\frac{E_{a}}{RT}+\alpha \ln C_{A}$$

Consequently, the kinetic parameters for *k*_1obs_, *k*_2obs_, and *k*_3obs_ can be calculated employing multiple linear regression and using the data from Table 2 and Table 3. Multiple linear regression when applied gives satisfactory goodness of fit as *R*^2^ ≥ 0.85 (Table 5 and Appendix A). The objective reaction (xylose dehydration to furfural) has higher activation energy than the crucial side reaction (decomposition of xylose to humin). Nevertheless, the effect of catalyst concentration was more significant on xylose dehydration as compared to xylose decomposition, as revealed by the high α value for *k*_1obs_ than *k*_2obs_:(7)$$k_{1\mathrm{obs}}=8.91\times 10^{8}\exp\left(-\frac{81800}{RT}\right)C_{{\mathrm{CrCl}}_{3}\cdot {6H}_{2}O}^{1.35}$$
(8)$$k_{2\mathrm{obs}}=2.60\times 10^{7}\exp\left(-\frac{66500}{RT}\right)C_{{\mathrm{CrCl}}_{3}\cdot {6H}_{2}O}^{0.204}$$
(9)$$k_{3\mathrm{obs}}=7.13\times 10^{8}\exp\left(-\frac{93020}{RT}\right)C_{{\mathrm{CrCl}}_{3}\cdot {6H}_{2}O}^{0.268}$$

The comparison of kinetic results with published literature is appropriate but challenging because of discrepancies in experimental conditions. Generally, the observed rate constants are a function of temperature, pressure, and the reaction system. Activation energy data for xylose dehydration compiled in the Appendix A showed a great variation in values (76–152 kJ mol^−1^). Lamminpä ä et al., explored the kinetics of formic acid-catalyzed xylose dehydration and the calculated activation energies were 152 kJ mol^−1^ and 161 kJ mol^−1^ for *k*_1obs_ and *k*_2obs_, respectively [12]. The reported activation energies were almost two times higher than this system because the reaction conditions applied were entirely different. The applied xylose concentration was only 0.08–0.2 M, whereas the reaction temperature was 130–200 °C. Low substrate concentration and high reaction temperature generally increase the product selectivity in organic synthesis [48]. Similarly, the *E*_a_ determined by Chen et al., was 108.6 kJ mol^−1^ and 105 kJ mol^−1^ for *k*_1obs_ and *k*_2obs_, respectively, when acetic acid (0.5 M) was used in a high-pressure reactor [32]. It has been observed that the reaction medium plays a critical role in the reaction energy barrier [49]. The observed activation energy changes from 145 kJ mol^−1^ to 114 kJ mol^−1^ when the reaction medium was changed from the aqueous phase to the GVL phase under identical reaction conditions (H_2_SO_4_ catalyst and 145–175 °C). Henceforward, it can be concluded that system reactivity and selectivity dynamically demonstrate the activation energies in any reaction. For a systematic comparison, all the parameters must be identical; otherwise, activation energy comparison seems inappropriate.

## 3. Materials and Methods

### 3.1. Materials

Xylose (99.0%), *p*-toluenesulfonic acid (*p*TSA, 99.80%), oxalic acid (99.70%), maleic acid (99.70%), malonic acid (97.50%), and succinic acid (99.8.0%) were bought from Shanghai Aladdin Biotechnology Co., Ltd. (Shanghai, China). N,N-dimethylformamide (DMF, 99.5%), dimethyl sulfoxide (DMSO, 99.9%), isopropyl alcohol (IPA, 99.80%), and Lewis acids (CrCl_3_·6H_2_O, AlCl_3_, and NH_4_Cl) were obtained from a local supplier namely Beijing Chemical Works (Beijing, China). All the analytical standards such as furfural (99.9%), Lyxose (99.9%), xylulose (≥99.0%), and xylose (≥99.7%) were supplied by Shanghai Aladdin Biotechnology Co., Ltd. Water with very high resistivity (≥18.25 mega-Ohm (MΩ)) was obtained from the available laboratory purification system.

### 3.2. Conversion of Xylose to Furfural

The conversion experiment was performed similarly to the experimental procedure of conversion of C_6_ sugars to 5-hydroxymethylfurfural as defined in our previous publications [33,34]. Briefly, a 3-neck Pyrex glass round bottom flask equipped with a total reflux condenser was used as a reactor and 50 mL reaction volume was used. The predefined molar concentration of xylose and required amount of catalyst was added into the reactor and homogenized using a stirrer. Organic acids such as oxalic acid (*p*K_a1_ = 1.27), maleic acid (*p*K_a1_ = 1.90), malonic acid (*p*K_a1_ = 2.83), succinic acid (*p*K_a1_ = 4.20), and *p*TSA (*p*K_a1_ = −2.80) were used as catalysts, whereas, CrCl_3_·6H_2_O, AlCl_3_, and NH_4_Cl were used as Lewis acid co-catalysts. The process was optimized by employing different reaction parameters in order to achieve the maximum product yield. Sample collection was performed after a predetermined time interval using a syringe pump and samples were preserved in the refrigerator until analyzed.

## 4. Analytics

### 4.1. Product Quantification

Product concentrations were determined using a HPLC system (Shimadzu, Kyoto, Japan) equipped with an Aminex^®^ HPX-87H strong acid cation exchange resin column (300 mm × 7.8 mm, Bio-Rad, Hercules, CA, USA) at 65 °C with a RID-10A differential refractive index (RI) detector. The analysis was performed with 0.8 mL/min of M/200 sulfuric acid solution. Ultrapure water (≥18.25 MΩ) was used for the dilution of the samples and filtration was performed using 0.22-micron syringe filters. An auto-injection module was applied and 20 micro litter (µL) sample volume was injected for analysis.

Process efficiency was calculated based on xylose conversion (*X*_x_), furfural yield (*Y*_F_), furfural selectivity (*S*_F_), and carbon balance (CB). These parameters were calculated using Equations (10)–(13). CB was calculated in order to estimate the number of undesirable by-products and summarized as humin (HUM):(10)$$\mathrm{Xylose}\;\mathrm{conversion}\;(X_{X})=\left[1-\frac{\mathrm{Xylose}\;(M)\;}{\mathrm{Xylose}\;\mathrm{in}\;\mathrm{feed}\;(M)}\right]\times 100$$
(11)$$\mathrm{Furfural}\;\mathrm{yield}\;(Y_{F})=\left[\frac{\mathrm{Furfural}\;(M)\;}{\mathrm{Xylose}\;\mathrm{in}\;\mathrm{feed}\;(M)}\right]\times 100$$
(12)$$\mathrm{Furfural}\;\mathrm{selectivity}\;(S_{F})=\left[\frac{\mathrm{Furfural}\;\mathrm{yield}\;(Y_{F})}{\mathrm{Xylose}\;\mathrm{conversion}\;(X_{X})}\right]\times 100$$
(13)$$\mathrm{Carbon}\;\mathrm{balance}\;(CB)=\left(5\times X_{X}\right)-\left(5\times Y_{F}\right)=Y_{HUM}$$

### 4.2. Development of Kinetic Models

The conceivable reaction route of xylose conversion is outlined in Scheme 1 and the detailed reaction mechanism in Appendix A. Although different reaction routes for xylose dehydration have been under consideration for a long time; however, the reaction as described in Scheme 1 is mostly considered the principal route [10,27,29]. It has been observed that xylose conversion produces furfural and unwanted side products (HUM), simultaneously. Correspondingly, the decomposition of furfural into side products is also inevitable. The discrete reaction details are intricate and an exact mechanism is still not fully developed but the reactions (Equations (14)–(16)) can be used as simplified reaction pathways following the Scheme 1. Accordingly, the subsequent kinetic models could be established as:(14)$$\frac{dC_{X}}{dt}=-(k_{1\mathrm{obs}}+k_{2\mathrm{obs}})C_{X}$$
(15)$$\frac{dC_{F}}{dt}=k_{1\mathrm{obs}}C_{X}-\;k_{3\mathrm{obs}}C_{F}$$
(16)$$\frac{dC_{\mathrm{HUM}}}{dt}=k_{2\mathrm{obs}}C_{X}+k_{3\mathrm{obs}}C_{F}$$
where *k*_1obs_, and *k*_2obs_, are the reaction rate constants of the conversion of xylose into furfural and xylose decomposition to side products (humin), *k*_3obs_ is the decomposition rate of furfural to undesired products (humin). Humin (HUM) is the sum of condensation products and quantified using a material balance as defined in Equation (13) [50]. *C*_X_, *C*_F_, and *C*_HUM_ denote the molar concentrations of xylose, furfural, and humin, respectively.

### 4.3. Estimation of Kinetic Parameters

Experimental data was used for the evaluation of the kinetic parameters using MATLAB R2016a. The estimation of kinetic parameters such as rate constants (*k*) and activation energy (*E*_a_), was performed using numerical integration of Equations (14)–(16) by ode45 function similar to our previously published procedure [33,34]. The parametric optimization was achieved employing the least-squares method. The numerical calculation of the Arrhenius factor (*A*) and activation energy (*E*_a_) was done by MATLAB R2016a software employing the Arrhenius equation.

## 5. Conclusions

Catalytic transformation of xylose into furfural was elaborated over different organic acids in aqueous as well as in organic solvent medium. The reaction yield was optimized by navigating different reaction conditions and adding different chlorides as Lewis acid co-catalysts. An extensive range of reaction temperatures (110–160 °C) and Lewis acid concentrations (0.1–0.5 M) was applied for process optimization. Furfural yield and xylose conversion were compared based on experimental results and kinetic modeling. A kinetic model was developed comprising the reactions of dehydration of xylose to furfural, decomposition of xylose to humin, condensation of furfural to humin. Results illustrated the catalytic inferiority of pure *p*TSA in the aqueous medium; however, much improvement in furfural selectivity was achieved by adding various chlorides as Lewis acid co-catalysts. The addition of CrCl_3_·6H_2_O as a co-catalyst improved the reaction yield up to 30.1% at 100 °C in an aqueous medium. Replacement of the reaction medium from water to DMSO gives the maximum furfural yield of 53.1% using *p*TSA and CrCl_3_·6H_2_O in a molar ratio of 10:1 in 60 min at 120 °C. Modeling results illustrated the model accuracy with high *R*^2^. The model predicted values validated that dehydration of xylose to furfural (*k*_1obs_) and decomposition of xylose to humin (*k*_2obs_) are the key reactions. *k*_1obs_ and *k*_2obs_ were found to display similar oscillations, while condensation of furfural to humin (*k*_3obs_) was in the range of 10^−3^ to 10^−4^ with narrow vacillations contingent on the reaction temperature and chloride concentration. It was observed that *k*_1obs_ increased with an increase in temperature and decreased with an increase in Lewis acid concentration. However, *k*_2obs_ increased with an increase in temperature and displays a narrow oscillation with an increase in Lewis acid concentration. The activation energies for xylose dehydration, xylose decomposition, and condensation of furfural were determined to be 81.8, 66.5, and 93.0 kJ/mol, respectively. It was also observed that xylose dehydration to furfural was more sensitive toward Lewis acid concentrating than xylose decomposition and furfural condensation to humin. The obtained results further suggest that furfural selectivity can be improved by depressing the rate of xylose condensation.

## Data Availability

The data presented in this study are available in the article and Appendix A.

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
