# Peer review of "Catalytic Conversion of Xylose to Furfural by p-Toluenesulfonic Acid (pTSA) and Chlorides: Process Optimization and Kinetic Modeling"

_molecules, 2021, doi:10.3390/molecules26082208_

Round 1
Reviewer 1 Report
The manuscript entitled "Catalytic conversion of xylose to furfural by p-toluenesulfonic acid (pTSA) and chlorides: Process optimization and kinetic modeling" authored by Sajid et al., presented an advanced optimization of the furfural production process with the use of a combination of organic acids and chlorides with organic solvents. The article is written correctly and clearly. However there are some lacks, so I have a few suggestions that clarify some issues:
- I think it is worth considering the use of the words "eco-friendly" or "ecological" in relation to a reaction system in which toxic organic and/or inorganic compounds are present. It is well known that xylose can be enzymatically converted, so it would be worth mentioning in the introduction why chemical catalysts were chosen, especially in the context of being environmentally friendly. I think it would be appropriate to read the following articles (https://doi.org/10.1016/j.bioorg.2019.01.043; https://doi.org/10.3390/ma12193167).
- Also, I think there is definitely not enough information in the "Experimental" section. Due to the variability of the conditions in the presented research, in my opinion it would be good to extend this section and describe the contents of the tested reaction systems.
- What was the reason for selecting the initial concentrations of xylose, pTSA and different Lewis acids during xylose conversion in water as solvent?
- The authors correctly noticed that the highest yields of furfural production were obtained with the use of different concentrations of CrCl3 depending on the reaction medium (water or DMSO). However, the article does not attempt to explain this result.
- Moreover the results of the efficiency of xylose conversion with the use of various organic acids are also presented, however, again no attempt has been made to explain why pTSA allows for furfural productivity to be higher than the other acids.
Reviewer 2 Report
This paper attempts to study the effects of Lewis and Brønsted acidities on production of furfural from xylose and suggests different solvents for reaction, and changing reaction conditions in order to study the kinetic of the reaction. The manuscript is well structured and generally clear in its purpose. The authors described the kinetic aspect of the reaction very well and in details. However, I think the manuscript can be accepted for publication after addressing the following comments:
- Recent studies attempt to demonstrate green approach for biomass valorization process. Although, using water as the solvent had a weaker result compered to organics one, Anyway, the aim of recent studies is to optimize the catalysts and reaction condition in the presence of green solvent. Do you think the best solvent of your work (DMSO) can be considered as a green and also economic option?
- Regarding the catalyst, I believe that recent researches are moving on to use a heterogeneous catalyst having bifunctional acidities. But in this research, two separated homogeneous catalysts were used. Do you think it could be a good selection in the aspect of reactor corrosion, recovery and the possibility to use them in an industrial scale?
- Considering that the catalyst is homogeneous and the solvent is non-green, is it the performance of catalyst high respect to literature?
Best regards,
Round 2
Reviewer 1 Report
I'm happy to see that authors improved manuscript due to previously marked comments/remarks. In my opinion it can be consider to publish in current state.